# COVID-19 Lockdown and Mental Health in a Sample Population in Spain: The Role of Self-Compassion

**DOI:** 10.3390/ijerph18042103

**Published:** 2021-02-21

**Authors:** María Elena Gutiérrez-Hernández, Luisa Fernanda Fanjul, Alicia Díaz-Megolla, Pablo Reyes-Hurtado, Jonay Francisco Herrera-Rodríguez, María del Pilar Enjuto-Castellanos, Wenceslao Peñate

**Affiliations:** 1School of Medicine, Universidad de Las Palmas de Gran Canaria, 35001 Las Palmas, Spain; luisafernanda.fanjul@ulpgc.es; 2Servicio de Psiquiatría, Complejo Hospitalario Universitario Insular-Materno Infantil, 35016 Las Palmas, Spain; pabloreyeshurtado@gmail.com (P.R.-H.); jherrerarodriguez5@gmail.com (J.F.H.-R.); 3Departamento de Educación, Universidad de Las Palmas de Gran Canaria, 35001 Las Palmas, Spain; alicia.diaz@ulpgc.es; 4School of Valladolid, Universidad de Valladolid, 37008 Salamanca, Spain; piluca.enjuto@gmail.com; 5Departamento de Psicología Clínica, Psicobiología y Metodología, Universidad de La Laguna, 38200 Santa Cruz de Tenerife, Spain; wpenate@ull.edu.es; 6Instituto Universitario de Neurociencia, Universidad de La Laguna, 38200 Santa Cruz de Tenerife, Spain

**Keywords:** mental health, emotional disorders, self-compassion, COVID-19

## Abstract

Previous data support that mental health is affected during pandemic and lockdown situations. Yet, little is known about the positive factors that protect mental health during a lockdown. This study analyzed mental health status—particularly emotional problems—and the role of several sociodemographic and clinical variables; it also explored whether there is a positive relationship between self-compassion and better mental health status. A cross-sectional study was carried out in Spain with the participation of 917 fluent Spanish-speaking residents in a survey conducted approximately midway through the COVID-19 lockdown. The survey tested for anxiety, depression, and stress using the Depression Anxiety Stress Scale-21 (DASS-21), the Self-Compassion Scale (SCS) to measure self-compassion values, and the Perceived Vulnerability to Disease Questionnaire (PVDQ) to assess the degree of risk perceived by participants. Around 30% of the individuals surveyed (recruited by snowball sampling) showed clinically significant levels of anxiety, depression, and stress. The variables most frequently associated with anxiety, depression, and stress were low levels of self-compassion, age, gender, previous physical symptoms, a previous mental disorder, being a student, and perceived vulnerability to disease. We discuss the hypothetical protective role against anxiety, depression, and stress of certain skills such as self-compassion and the possibility that increasing self-compassion may be used to promote better mental health in similar situations.

## 1. Introduction

Four cases of atypical pneumonia caused by a coronavirus were brought to light in the Chinese city of Wuhan at the end of December 2019. The new coronavirus was assigned the name SARS-CoV-2 because of its similarity to the virus responsible for the outbreak of the severe acute respiratory syndrome (SARS) respiratory infection in 2002 and 2003 [1]. Between December 2019 and March 2020, coronavirus disease (COVID-19) spread quickly throughout Asia and Europe. In March 2020, cases were recorded in every continent and the WHO declared COVID-19 a pandemic. Imposed lockdown of the general population became widely used as a strategy for containing the spread of the disease. This lockdown of large masses of the population has had serious repercussions at all levels, affecting the health of the population and the economy and social life in general [2].

The epidemiological, clinical, and therapeutic aspects of COVID-19 were all given emergency status, with a significant scientific response triggered around the world to find ways to stop the pandemic, save as many lives as possible, and prevent new outbreaks. However, there are harmful social contexts with the property of significantly altering normal life. These alterations also affect mental health. The effects of the pandemic on mental health, both of the frontline health workers actively engaged in the struggle against the disease and of the general population under lockdown, have also been a major area of research as some of the most significant side effects of COVID-19 [3]. Several studies [4,5,6,7,8,9,10,11,12,13,14] have reported a considerable rise in adult mental disorders during the COVID-19 lockdown; children’s psychological well-being has also been found to be seriously affected [15].

The mental disorders most frequently reported are anxiety and depression. Depression is a leading cause of disability and has a major impact on overall health worldwide, leading to suicide and self-harm in its most extreme cases (World Health Organization, n.d.). Anxiety is also widespread and is the sixth major cause of illness [16]. The fear generated by the pandemic and the lockdown situation may exert a specific pressure on the emotional distress of many people, over and above the stress usually generated by fear and lockdown [3]. Even on a lesser scale, it may affect individuals’ mental state, causing them to dwell obsessively on the fearful possibilities of infection, making them irritable and causing loss of sleep or insomnia [17].

Researchers have also explored other extrinsic factors in addition to emotions that can predispose or protect individuals against anxiety and depression, mostly sociodemographic factors (e.g., isolation, characteristics, and physical conditions of the housing in which lockdown was spent) and previous medical conditions. Of all these, isolation and lessened social interaction have been pinpointed as critical elements in spurring or worsening emotional problems, independently of the age group involved [18,19].

Brooks et al. (2020) reported the pivotal role of isolation, often associated with negative psychological effects, and considered fear of and obsession with the virus as the most frequent variable in lockdown [20,21]. Because those contexts are harmful by themselves and cannot be reinterpreted, psychological processes based on acceptance could be a useful strategy to cope with mental problems. However, the studies conducted have seldom analyzed the positive psychological variables that allow people to be resilient. From a therapeutical point of view, there are several psychological mechanisms and processes that regulate emotional distress. Cognitive behavior analysis makes it possible to explore the cognitive changes caused by emotional problems. It eliminates the cognitive distortions associated with emotional distress through cognitive restructuring. In the current situation, in which the virus and the lockdown are real, however, individuals’ thought processes cannot be considered to constitute a distortion of reality [22]. Taking this into account, acceptance processes—related to third-generation therapies such as acceptance and commitment therapy (ACT), mindfulness, and self-compassion—may work better to relieve distressing emotional states. Because these critical situations (i.e., the pandemic and the lockdown) are harmful by themselves, painful emotions are part of their consequences on human life. The common factor in third-generation therapies is the acceptance of emotional suffering. Acceptance therapies can teach people to become aware of their emotions and learn to relate to them in a more balanced way [23]. Self-compassion may be a resource to promote acceptance. Considering this, the present study explored mental health problems during the COVID-19 pandemic, their sociodemographic and psychological predictive factors, and particularly whether self-compassion played a significant role among these factors.

Self-compassion, defined as the ability to treat suffering sympathetically and with the awareness that it is part of being human, recognizes that suffering and our imperfection are part of the human experience. It involves being touched by and open to one’s own suffering, not avoiding or disconnecting from it [24].

Self-compassion has been associated with lower levels of emotional distress [25,26]. In fact, training in self-compassion has proved to be effective in reducing emotional distress [27,28,29]. Moreover, several systematic reviews and meta-analyses have evidenced that practicing mindfulness and self-compassion is also effective for other mental problems and disorders [30,31,32,33,34,35,36].

In light of previous studies, the main aim of the present research was to assess emotional distress during the COVID-19 lockdown, measuring anxiety, depression, and stress levels. We also intended to explore the relationship between sociodemographic, clinical, and psychological variables, on one hand, and the level of emotional distress, on the other. The following sociodemographic variables were taken into account: age, gender, academic background, employment status, physical distress symptoms, any previous mental disorders, whether individuals spent the lockdown alone or accompanied and with children or not, previous experience with meditation, and the physical characteristics of the housing where the lockdown was spent. The psychological variables assessed were fear of infection (i.e., perceived vulnerability to the disease) and self-compassion. These variables were measured to identify possible predictive variables of the presence of clinical levels of anxiety, depression, and stress.

## 2. Materials and Methods

### 2.1. Participants

A non-representative convenience sample was used. The sample size was 917 individuals, mostly women (71.8%). The average age of participants was 42.55 years old (SD = 14.29). The total group ranged between 18 years old and 86 years old. Regarding academic background, 71% of participants had university studies, 22% had secondary studies, and the remaining 7% had primary studies. Almost half of the sample group was engaged in active employment (28.52% participants were teleworking, 12.29% worked outside the home, and 16.91% were healthcare workers), 12.18% were students, and the rest were in various states of inactivity (unemployed, furloughed, on sick leave, permanently disabled or retired).

Inclusion criteria were being a resident in Spain during the lockdown, being an adult (18 years old or older), having at least a primary education level, and having a good understanding of the Spanish language (reading).

### 2.2. Materials

Form. A Google form was used to collect information on the variables gender, academic background, employment status, existence of physical symptoms in the two weeks before the study, previous history of mental disorders, whether the lockdown was spent alone or not, whether children were present, previous experience with meditation or not, and the characteristics of the housing where the lockdown was spent. The form included the following inventories:

Depression Anxiety Stress Scale-21 (DASS-21) [37]. This is a self-applied questionnaire with three subscales (i.e., stress, depression, and anxiety) composed of seven items each. Responses are given on a four-point Likert scale ranging from 0 (nothing) to 3 (a lot). Scores on each subscale range from 0 to 21. The version used was a validated Spanish translation [38] with a total alpha coefficient of 0.96 and an equally high alpha coefficient for the depression (0.93), anxiety (0.86), and stress (0.91) subscales. Overall, the scales in the original English and the Spanish version have sound psychometric properties [39,40]. Our sample had a total Cronbach alpha index of 0.94, and the subscales had the following values: depression (0.85), anxiety (0.83), and stress (0.90).

Perceived Vulnerability to Disease Questionnaire (PVDQ) [41]. This is also a self-reported questionnaire made up of two subscales: perceived infectability (seven items) and germ aversion (eight items). Responses are provided on an eight-point Likert scale ranging from 1 (totally disagree) to 7 (totally agree). We used the validated Spanish version [42], which had a total Cronbach alpha index of 0.69 and alpha indices of 0.78 for the perceived infectability subscale and 0.59 for the germ aversion subscale. The validated Spanish version had excellent test-retest reliability (r = 0.95 for the perceived infectability subscale and r = 0.98 for the germ aversion subscale). Our sample had a total Cronbach alpha index of 0.76; the alpha indices of the perceived infectability and germ aversion subscales were 0.85 and 0.66, respectively.

Self-Compassion Scale (SCS) [43]. This is a questionnaire covering three dimensions: Self-Kindness versus Self-Judgment, Common Humanity versus Isolation, and Mindfulness versus Over-Identification. There is a short version of the original scale [44] with sound psychometric qualities [45]. The short version is made up of 12 items with responses measured on a five-point (5) Likert scale ranging from 1 (hardly ever) to 5 (almost always). We used the validated Spanish version of the short version of the SCS [46], which has a Cronbach alpha index of 0.85. Our sample had a Cronbach alpha index of 0.84.

### 2.3. Design

We used a cross-sectional design and recruitment was conducted by online snowball sampling. Prospective participants were contacted through social media and asked to recruit other participants. The target population was people residing in Spain during the time period of the study, from 14 April 2020 to 21 April 2020 (lockdown was imposed on 13 March 2020 and lifted on 15 May 2020, so the study took place approximately halfway through). The lockdown was strict throughout Spain with people only allowed to leave their homes to go to work, buy medicine or staple goods, or attend to emergencies. All non-essential establishments were closed, including schools. The variables considered as foreseeably detectable were depression, anxiety, and stress. The variables used to tally the results were age, gender, academic background, work status, physical symptoms of the virus in the two weeks prior to the study, previous history of mental disorders, whether the lockdown was spent alone or not, whether children were present or not, previous experience with meditation, the characteristics of the housing where the lockdown was spent, fear of catching the virus, and levels of self-compassion.

### 2.4. Procedure

The questionnaires were made available to the target sample (i.e., fluent Spanish-speakers who were residents in Spain at the time of the lockdown) by the researchers between 14 April 2020 and 21 April 2020. Researchers’ contacts were then asked to forward the questionnaires to their own families and acquaintances. The completed forms were returned before 21 April 2020. All questionnaires were self-applied and anonymous to respect confidentiality.

### 2.5. Ethics

The project was approved by the Medical Research Ethics Committee of the university hospital “Complejo Hospitalario Universitario Insular Materno-Infantil.” The surveys contained information about the goals of the study for participants to read. All participants gave their signed informed consent in compliance with the Spanish Data Protection Act and in line with the rights contemplated under the Declaration of Helsinki.

### 2.6. Data Analysis

R Core Team 2020 statistics software (R Foundation for Statistical Computing, Vienna, Austria) was used to process the results. Descriptive data were calculated with frequency analysis. Logistic regression models were used to chart the variables depression, anxiety, and stress. Cases and non-cases were determined according to the presence of clinically significant scores. Specifically, cases were considered if participants were included in slight, moderate, serious, or severe categories. A univariate and direct regression method was used. All coefficients with a probability equal to or less than 0.05 were considered statistically significant.

## 3. Results

First, social and clinical sample characteristics were described. These data are summarized in Table 1. Most participants reported not having any previous mental disorders and living in apartment buildings with more than one adult. Moreover, around one-third of participants experienced physical symptoms in the previous two weeks, were living with children and had some experience with meditation.

The next step was to analyze the presence of emotional disorders. The three categories of the DASS-21 (depression, anxiety, and stress subscales) were used. The average scores were 5.24 for anxiety (SD = 6.29), 7.52 for depression (SD = 7.14), and 11.29 for stress (SD = 8.6). As shown in Table 2, the percentage of clinically significant levels amounted to 30% over the whole sample group. Clinical significance was considered when participants had a non-normal score in each subscale (i.e., slight, moderate, serious, or severe levels).

The average result for the levels of self-compassion as measured by the SCS was 19.8 (over a range of 6–30) with a standard deviation of 4.4. The classification given to the sample results was LOW (for results below the average minus the standard deviation), NORMAL (for results equal to or above the average, factoring in the standard upper and lower deviations), and HIGH (for results above the average plus the standard deviation). Specifically, 8.07% were low, 44.93% were normal, and 47% were classified as higher than average in self-compassion. As regards the results of perceived vulnerability to disease (measured on the PVDQ),53.22% of participants were in the clinical range of germ aversion while only 20.5% were in the clinical range of perceived infectability.

Once the levels of anxiety, depression, and stress had been described, they were cross-referenced to the psychosocial and socio-demographic profiles to observe which, if any, of the variables bore an association with the mental state of the participant in lockdown. Three logistic regression models were designed to predict the anxiety, depression, and stress variables according to the profiles described above with the group split between individuals who did not show clinical levels and individuals who did, in line with the data already presented in Table 2. Table 3, Table 4 and Table 5 show the results of this analysis.

Table 3 shows the results of the logistic regression analysis for the anxiety variable. As can be observed from the data, seven variables were considered to be significant predictors of anxiety—self-compassion, perceived vulnerability to disease, age, gender, previous experience of physical symptoms, history of previous mental disorders and being a student. Self-compassion levels were inversely related to anxiety levels, that is, participants without anxiety had higher self-compassion scores (OR < 1), and perceived vulnerability to disease was positively related to anxiety levels (OR > 1). Cases of anxiety were also predicted by age (i.e., there was a higher incidence in younger participants), gender (i.e., women), previous experience of physical symptoms, previous history of mental disorders, and the fact of being a student.

The predictive results for cases of depression (Table 4) were almost exactly the same for many variables except for the variables of secondary school studies and passive work status (i.e., on sick leave, permanently disabled, retired, furloughed, or unemployed).

The prediction of cases of stress (Table 5) followed a similar pattern to the trends in anxiety and depression but introduced two new variants—there were more cases of stress in people who lived with someone else or who shared their homes with children than in those who did not.

## 4. Discussion

The present study had two objectives, the first of which was to assess the levels of anxiety, depression, and stress in the local adult population during the COVID-19 lockdown. The second objective of the study was to explore which psychological factors and sociodemographic factors may be associated with these changes. The overall results showed that around 30% of participants reached clinical levels of anxiety, stress, and depression during the lockdown. The study also allowed us to identify predictive sociodemographic and psychological factors.

The clinical levels of anxiety, depression, and stress found in around one-third of the participants are broadly in line with other similar studies [4,8,13,47]. The bulk of these studies were conducted in China but there are also similar results from studies in Bangladesh [5], India [7], Italy [9], Vietnam [10], United Kingdom [12], and Spain [11,17].

Age and gender were the two main variables considered in all these studies. This study revealed that younger participants exhibited higher psychological distress, a result that is also in line with most published studies [48,49,50,51,52]. The reasons for this greater suffering among younger people may be that the crisis produced uncertainty about their academic and career possibilities (this is the second crisis that they face, the first being the economic crisis in 2008), and the fact that they are more dependent on social networks for their information and support, which may substantially increase the distress.

Men were found to be less psychologically distressed than women, also a recurrent feature found by most other studies [3,17,49,53]. This greater distress experienced by women may be accountable to their general greater psychological vulnerability to traumatic and adverse situations [54]. Having secondary or higher studies was associated with a higher level of depression, also consistent with other studies [47,55,56,57,58]. This may be because these participant groups have a greater awareness of the real health risks for themselves and their families and of the socioeconomic repercussions of the pandemic. In younger participants, being a student was linked to more symptoms of anxiety, depression, and stress, in line with other studies [5,13,17,51,59,60]. As noted previously, this may be due to the greater uncertainty generated by the pandemic on their future academic and career aspirations.

As regards health, the fact of having experienced physical symptoms in the weeks prior to the self-report also influenced the variables. This result is consistent with those of other studies [13,61,62]. Participants may have associated some of the physical symptoms with a possible SARS-CoV-2 infection. Additionally, the existence of a previous history of mental disorders was associated with higher levels of anxiety, depression, and stress [18,63,64,65,66,67,68]. In this regard, the pandemic and the lockdown may be stressful factors that produce instability in a previous mental condition [69,70,71].

In the area of social and work relations, it was logical to expect work status to be associated with the level of depression. There were more cases of depression among participants who were not engaged in active employment (i.e., people on sick leave, disabled, retired, unemployed, or furloughed). This is a constant in the literature on anxiety, depression, and stress [7,9,17,59]. People not actively engaged in employment under lockdown experience greater isolation, more inactivity, and usually have a lower income, all of which are factors that increase symptoms of depression.

Although the onset of emotional disorders may stem from an initial affective vulnerability [72], the experience of lockdown with children was associated with higher levels of stress, as found in other studies [3,12,51,72,73]. Meeting children’s needs in lockdown is likely to be a trigger for stress. It was surprising, however, to find that living without other adults around was associated with lower levels of stress, anxiety, and depression [51]. The family network and social interaction with its members are usually looked upon as a predictive factor of stress. Therefore, it may well be that the conditions of lockdown generate stressful family situations, turning what would normally be a hypothetical protective factor into a possible risk factor. Future research should be directed at analyzing the “other adult” component according to categories, such as parents, partners, and friends [55].

Another unexpected result was the fact that health workers did not report greater levels of stress, anxiety, or depression. This is commonly found in studies of this nature [61,70,74]. This apparently contradictory finding would probably be clarified by a more in-depth analysis of the type of health workers, and especially of their direct contact with patients with COVID-19, as has been reported by other studies of this type [4].

Perceived vulnerability to disease and levels of self-compassion, as was to be expected, were associated with levels of anxiety, depression, and stress. Perceived Infectability was associated with greater concern about COVID-19 [58] and heightened fear of infection, thereby constituting a risk variable that affects overall mental health [75] and specifically anxiety and depression [76,77] and stress.

Levels of self-compassion were related to lower levels of anxiety, depression, and stress in general and seemed to work as a protective factor. These results are consistent with those of previous studies that also used the DASS-21 scores [34]. People with higher levels of self-compassion react to adverse events with better emotional regulation [35].

The role played by self-compassion in resisting stress, anxiety, and depression during the COVID-19 pandemic and lockdown has not received the same depth of analysis as the other variables. Yet, when it has been explored, it has been found to be related to positive reaction and greater emotional balance [73,78].

There is solid empirical evidence that positive levels of self-compassion improve mental health [32,33,34,35,79,80,81,82] and lessen the negative consequences of psychological parameters such as stress, anxiety, and depression [28,31,34,83,84,85,86,87,88,89]. A review of the literature on epidemics that occurred in the last two decades shows that compassion is a positive strategy to deal with the negative impacts of these diseases [61]. Similar data show that self-compassion also acts as a protective factor in high-stress situations [90,91]. Self-compassion reduces vulnerability particularly in the presence of emotional disorders; in fact, there is a general model that shows that mindfulness and self-compassion are protective mechanisms against mental illness [24,43]. The effectiveness of self-compassion in emotion regulation may be due to its different components. In this regard, we consider that self-kindness can promote a psychological state of well-being. Common humanity may reduce the feeling of loneliness generated by the lockdown. Finally, mindfulness helps to identify distress in order to accept it. Obviously, these hypothetical differential implications of the components of self-compassion should be tested in future experimental studies.

This study has several limitations and shortfalls. Specifically, the information was obtained through self-reports without any control over the level of honesty of participants. It was a cross-sectional study carried out at a specific moment during the lockdown (approximately halfway through) and therefore does not give us a global picture of the total psychological distress of enforced lockdown during the pandemic. Longitudinal studies are required to explore how these results evolve over time. The sample was a convenience or snowball sample recruited online. Thus, it was a broad-ranging sample that is not representative of any particular target group and may have underrepresented older age groups since older people tend to use fewer digital devices.

Nevertheless, if considered with due caution, the results may have significant practical implications. The sociodemographic variables related to worse emotional adjustment to the situation make it possible to identify profiles of people at higher risk of experiencing anxiety, stress, depression, and related disorders. In addition, perceived infectability is a vulnerability factor that creates a distortion, that is, an exaggeration of the degree of risk, and can therefore be approached using cognitive restructuring techniques. Self-compassion, as an emotion regulation strategy, can also be used to train specific acceptance skills, thus fostering emotional resilience.

Although public health resources cannot provide psychological assistance to the entire population, other more cost-effective short-term, group therapy and practical strategies could be implemented to reduce the levels of emotional distress. Mindful-self compassion fits those characteristics and this intervention program gathers scientific evidence of its benefits for emotional distress [51,52,53]. There is also the possibility of being taught online, which makes them viable at times such as lockdowns and quarantines. To generalize the effectiveness of self-compassion training, future studies could be conducted to examine the benefits of self-compassion for specific psychological impacts derived from catastrophic situations.

## 5. Conclusions

During the COVID-19 pandemic lockdown, one-third of the participants in our study reported high levels of anxiety, stress, and depression. The variables most frequently associated with anxiety, depression, and stress were levels of self-compassion, age, gender, previous physical symptoms, previous history of mental disorders, being a student, and perceived vulnerability to disease. The results may help to identify the most vulnerable profiles in this type of situation and offer support interventions that minimize the collateral effects.

## Figures and Tables

**Table 1 ijerph-18-02103-t001:** Social and clinical characteristics of the sample (n = 917).

Variables	Categories	Frequency	%
Previous history of mental disorders	No	853	93.02
	Yes	64	9.98
Physical symptoms in the previous two weeks	No	580	63.25
	Yes	337	36.75
Characteristics of the place of residence	Apartment/townhouse	771	84.08
	House with garden estate	146	15.92
Number of adults in lockdown	1	151	16.47
	More than one	766	83.53
Presence of children in lockdown with the participant	No	590	64.34
	Yes	327	35.66
Previous experience with meditation	No	583	63.58
	Yes	334	36.42

**Table 2 ijerph-18-02103-t002:** Percentages per category and variable using the Depression Anxiety Stress Scale-21 (DASS-21) (n = 917).

Clinical Level	Anxiety (%)	Depression (%)	Stress (%)
NORMAL	72.96	68.7	73.06
SLIGHT	6.65	13.96	10.14
MODERATE	12	12.1	9.05
SERIOUS	3.49	2.84	5.34
SEVERE	4.91	2.4	2.4

**Table 3 ijerph-18-02103-t003:** Data of the univariate logistic regression analysis performed to predict the anxiety variable.

Variable	OR	95% CI	*p*-Value
Self-Compassion	0.84	0.8–0.87	<**0.001**
Germ Aversion	1.27	0.94–1.71	0.118
Perceived Infectability	1.93	1.36–2.73	<**0.001**
Age	0.99	0.98–1.0	**0.012**
Gender: Male	0.51	0.36–0.73	<**0.001**
Level of studies: Degree or Higher Studies	1 (ref)		
Secondary	1.11	0.77–1.59	0.56
Primary	0.93	0.48–1.71	0.828
Physical Symptoms: Yes	2.79	2.06–3.7	<**0.001**
Previous Mental Disorders: Yes	2.92	1.71–4.97	<**0.001**
No. of Adults in Lockdown: Alone	0.69	0.45–1.04	0.087
No. of Children: ≥1	1.15	0.84–1.56	0.387
Housing: House with Garden Estate	0.86	0.56–1.3	0.493
Meditation	1.16	0.85–1.57	0.343
Work Status: Health Worker	1 (ref)		
Student	1.99	1.16–3.34	**0.013**
On Sick Leave/Permanently Disabled/Retired	1.20	0.696–2.081	0.506
Teleworking	1.12	0.693–1.822	0.651
Working Outside the Home	1.30	0.735–2.274	0.369
Unemployed or Furloughed	1.09	0.647–1.829	0.756

Abbreviations: OR = odds ratio; CI = confidence interval; *p* = probability; ref = is the variable that is used as a comparator for the rest of the variables. Statistically significant *p*-Values are in bold font.

**Table 4 ijerph-18-02103-t004:** Data of the univariate logistic regression analysis performed to predict the depression variable.

Variable	OR	95% CI	*p*-Value
Self-Compassion	0.76	0.72–0.79	<**0.001**
Germ Aversion	1.08	0.81–1.44	0.586
Perceived Infectability	1.42	1–1.99	**0.046**
Age	0.98	0.97–0.99	<**0.001**
Gender: Male	0.58	0.41–0.8	**0.001**
Level of studies: Degree or Higher Studies	1 (ref)		
Secondary	1.66	1.18–2.32	**0.003**
Primary	1.42	0.74–2.5	0.232
Physical Symptoms: Yes	2.84	2.12–3.82	<**0.001**
Previous Mental Disorders: Yes	4.51	2.63–7.96	<**0.001**
No. of Adults in Lockdown: Alone	0.96	0.65–1.54	0.835
No. of Children: ≥1	1.25	0.93–1.67	0.14
Housing: House with Garden Estate	0.8	0.53–1.18	0.273
Meditation	0.85	0.63–1.14	0.285
Work Status: Health Worker	1 (ref)		
Student	5.17	3–9.12	<**0.001**
On Sick Leave/Permanently Disabled/Retired	2	1.16–3.49	**0.014**
Teleworking	1.32	0.8–2.22	0.279
Working Outside the Home	1.96	1.11–3.48	**0.021**
Unemployed or Furloughed	2.33	1.4–3.93	**0.001**

Abbreviations: OR = odds ratio; CI = confidence interval; *p* = probability; ref = is the variable that is used as a comparator for the rest of the variables. Statistically significant *p*-Values are in bold font.

**Table 5 ijerph-18-02103-t005:** Data of the univariate logistic regression analysis performed to predict the stress variable.

Variable	OR	95% CI	*p*-Value
Self-Compassion	0.84	0.77–0.83	<**0.001**
Germ Aversion	1.14	0.85–1.54	0.385
Perceived Infectability	2.21	1.56–3.1	<**0.001**
Age	0.97	0.96–0.98	<**0.001**
Gender: Male	0.38	0.26–0.55	<**0.001**
Level of studies: Degree or Higher Studies	1 (ref)		
Secondary	1.13	0.79–1.61	0.496
Primary	1.02	0.53–1.85	0.96
Physical Symptoms: Yes	2.76	2.04–3.75	<**0.001**
Previous Mental Disorders: Yes	2.3	1.34–3.91	**0.002**
No. of Adults Confined: Alone	0.65	0.42–0.98	**0.045**
No. of Children: ≥1	1.46	1.07–1.97	**0.015**
Housing: House with Garden Estate	1.05	0.7–1.56	0.806
Meditation	0.9	0.66–1.23	0.524
Work Status: Health Worker	1 (ref)		
Student	2.42	1.42–4.16	**0.001**
On Sick Leave/Permanently Disabled/Retired	0.89	0.5–1.57	0.684
Teleworking	0.94	0.58–1.55	0.817
Working Outside the Home	1.54	0.89–2.68	0.125
Unemployed or Furloughed	1.39	0.84–2.31	0.204

Abbreviations: OR = odds ratio; CI = confidence interval; p = probability; ref = is the variable that is used as a comparator for the rest of the variables. Statistically significant *p*-Values are in bold font.

## Data Availability

Data Availability Statements can be found at https://www.mdpi.com/ethics.

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
