# Peer review of "COVID-19 Lockdown and Mental Health in a Sample Population in Spain: The Role of Self-Compassion"

_ijerph, 2021, doi:10.3390/ijerph18042103_

Round 1

Reviewer 1 Report

This is an interesting study looking at the relationship between self-compassion and anxiety, stress depressive symptoms during lockdown in Spain.  I think this is a really interesting study and I like the ms. I have included some comments and questions that I hope the responses to will further strengthen the ms.

My main feedback is that in its current format self-compassion does not feel like the focus of the study. Self-compassion is written about towards the end of the introduction and discussion meaning it feels more of an afterthought rather than the purpose of the paper.

My other comment is that the data is cross-sectional and using phrases like ‘The impact of’ when there is no pre pandemic data available is unfounded. There is a potential that this sample would have had high DASS scores pre-lockdown. I have similar thoughts around ‘the protective role of self-compassion’. The study appears to be looking at associations between these constructs.

There are a number of formatting issues in the ms the formatting of the tables. Text appears within the headings making it difficult to read the tables. It may have also affected the start of the discussion as there is no Discussion heading and the text seems to start mid-sentence.

Other points as they occur throughout the document.

  1. P1 L21 self-compassion acts as a protective factor against serious mental disorders

Can these claims be made from the data? Study is cross-sectional and was mental health diagnosis assessed?

  1. P1 L30 the fact of being a student
  2. P2 Line 59- sentence unclear currently. Could be changed to ‘ The mental disorders most frequently reported are

anxiety and depression.’

  1. L60 depression is a leading cause of disability rather than disabling medical conditions
  2. L81 culled- should this be collated/ collected or something similar?
  3. L93 self-compassion may reduce rather than prevent depression/anxiety
  4. How were people identified/ recruited? Was the study advertised online in the first instance?
  5. Do the authors have a sense of whether or not their sample is representative of the population of Spain?
  6. L127 characteristics of the place of confinement
  7. Scs description. I would use the names of the subscales (i.e self-kindness vs self-judgment) rather than renaming them
  8. L158 I’m not sure what is meant by cross-survey -cross-sectional?
  9. Table 1 discussion indicates that the table contains sample characteristics, but I can’t see any of these details in the Table.

There appears to be a glitch in the formatting of the tables, so the info may be there, but not legible to me.

  1. Typical deviation (e.g. L232) – change to standard deviation
  2. How do dep/anx score compared to non-pandemic?
  3. L234 sentence about % clinically significant levels could be clarified (or maybe discuss 3 variables separately as that’s how they’re reported?).
  4. What is the range for DASS subscales? How do the findings compare to national levels previous studies?
  5. Given the impact of the pandemic/ lockdown on younger females were the data assessed for gender and age differences? Similarly, was self-compassion assessed for gender/age differences?
  6. I’m not clear whether the regressions presented were run with each variable individually or if all variables were entered together?

Similarly, given gender and age appear to be associated with higher levels of stress, anxiety, depression were these variables controlled for in the analysis?

  1. Table 3 the CI’s and OR moves between 2 and 3 numbers after decimal points.
  2. What significance level was set in the study? Could the authors indicate in the tables which variables were significant please.
  3. Table 5 Univariate is misspelt
  4. L381 ‘local adult population’ was area of Spain that participant’s resided in recorded too?
  5. L383 ‘people experience higher clinical percentages over the three variables (anxiety, stress and depression) during lockdown’ I’m not sure this claim can be made from the data as no pre lockdown data was recorded.

I’m also unclear if this means that the authors suspect people had higher levels of anx/dep/stress or if a higher proportion of people were experiencing these concerns.

  1. In light of the importance of looking at the SCS subscales individually, is this something the authors would investigate in future studies?
  2. This looks like a really interesting dataset and I wondered if the authors had considered conducting analyses looking at mechanisms?
  3. Given the focus of the study I feel the ms would really benefit from a more detailed discussion around the self-compassion. It currently feels a bit lost and occurs near the end of the document.
  4. Also, what are implications of the study? Why is this study important? Maybe including something around compassion being learnable? What future studies would the authors like to see?

I hope the authors find the comments useful in revising their ms.

Reviewer 2 Report

General comments

-The present study aims to investigate the impact of COVID-19 lockdown on the mental health of a Spanish population and to evaluate which factors could predict emotional distress. Although the self-compassion should be the most important psychological variable take into account, I think that the authors could give greater emphasis to this construct.  Therefore, I suggest to improve both introduction and discussion deepening this topic and the association with distress. Moreover, I suggest to deepen the therapeutic implications by adding them to the conclusions.

Specific comments

-The first sentence of the abstract more like a concluding sentence. Please modify it.

Line 59- please modify “emotional” in “emotional distress”.

-Please, formulate the aims in a clearer and more precise way.

-I suggest to specify if there are exclusion criteria based on age and education level.

-I would suggest to insert the paragraph of data analysis after the ethics one

-I would suggest to move the list of collected socio-demographic variables before the instrument description.

-The author declared:” The socio-demographic variables presented in Table 1 show that the sample was 187 mainly made up of women (71.8%)”, but this information is not present in Table 1. Please clarify this point.

-The author declared:” the percentage of clinically significant levels amounted to 30% over the whole sample group” and also “Three models of logistic regression were designed to predict the variables of anxiety, depression and stress according to said profiles with the group split between those who did not show clinical levels and those who did”, but is not clear which is the cut-off. Please clarify this point and specify the number of participants belonging to the two groups.

-The description of logistic regressions is not sufficient. Did the authors insert the variables in one block? Please, add the description of the model in more detail.

-Please insert the paragraph “Discussion”.

Line 391- Please see also other studies such as Castelli et al. 2020, Di Tella et al.,2020.

Reviewer 3 Report

This study approaches the impact that restrictions on mobility declared as a consequence of the COVID-19 pandemic caused on mental health in the Spanish population. It is very important to clinical practice, as they found that around a third of the participants reported high levels of anxiety, stress and depression. This impact on mental health was associated mainly with greater perceived vulnerability to illness, less self-compassion, younger age, female gender, being a student or unemployed, and having pathological physical or psychological antecedents. The study is well carried out. The reasons for the research are well described and the methodology is appropriate for the objectives posed. In addition, the sample was rather large (917 participants).

The publication of this study would provide very significant knowledge on a subject which is currently of strong worldwide concern.

The authors are suggested to: 1) specify where the discussion section begins, as there seems to have been a problem in the pdf generated by the platform (for example lines 313, 331, 380…), and 2) specify some psychotherapeutic techniques that could promote acceptance and self-compassion in regard to COVID-19, in particular, enlarge on the idea that appears in lines 479-480.

Round 2

Reviewer 1 Report

Many thanks to the authors for being so responsive to my comments.

I have a couple of minor suggestions:

I appreciate the authors amending the title on my feedback.

In its current form, it should be ‘The association between COVID-19 lockdown and the mental health of a sample population in Spain: The role of self-compassion’

Alternatively, the title could be ‘COVID-19 lockdown and mental health in a sample population in Spain: The role of self-compassion’

L19 : ‘Previous data support mental health’  should this be: Previous data supports that mental health…

I really appreciate the extra information in the introduction. I now feel the intro provides a strong rationale for the study.

I really liked the authors’ original first paragraph introducing the pandemic and I think L43 should be the opening line.

I’ve made suggestions for integrating L38-42 throughout the intro which might help the intro flow better.

Move L38-L39 “There are harmful social contexts with the property of significantly altering normal life. This alteration also affects mental health.”  to L53 after ‘prevent outbreaks recurring.’ Start sentence ‘However, there are harmful social contexts…)

L39 “Because those contexts are harmful by themselves and cannot be reinterpreted, psychological processes based on acceptance could be a useful strategy to cope with mental problems.” Move to 71 after [20,21].

L39/40 “Self-compassion can be resource to promote acceptance. In this sense, this study deals with the mental health problems in pandemic time, its sociodemographic and psychological predictive factors, and, especially, if self-compassion plays a significant role among those factors.” Move to L81 After [22]. As I think this is a good introduction to SC.

L60- ref 16 is formatted differently

L90 ‘In light to’ should be ‘In light of’

Thanks again.

Reviewer 2 Report

The authors made the required revisions and therefore the manuscript now appears clearer.

Author Response

Thank you for your precise and useful review. It is an example for us how to review manuscript.

Reviewer 3 Report

The authors have addressed all the comments and revised the manuscript accordingly

Author Response

(The authors gave the same response as above.)
